# Wildlife and Bait Density Monitoring to Describe the Effectiveness of a Rabies Vaccination Program in Foxes

**DOI:** 10.3390/tropicalmed5010032

**Published:** 2020-02-21

**Authors:** Paolo Tizzani, Angela Fanelli, Carsten Potzsch, Joerg Henning, Srdjan Šašić, Paolo Viviani, Mevlida Hrapović

**Affiliations:** 1Department Veterinary Science, University of Turin, V. Leonardo da Vinci 44, 10095 Grugliasco (TO), Italy; angela.fanelli@unito.it; 2Consultant Veterinary Epidemiologist, 16866 Tramnitz, Germany; carsten@potzsch.eu; 3School of Veterinary Science, The University of Queensland, Gatton, Qld 4343, Australia; j.henning@uq.edu.au; 4Wildlife Consultant, 81000 Podgorica, Montenegro; svaf61@gmail.com; 5Veterinary Consultant, 06123 Perugia, Italy; md2703@mclink.it; 6Veterinary Administration, 81000 Podgorica, Montenegro; mevlida.hrapovic@vu.gov.me

**Keywords:** rabies, vaccination campaign, Montenegro, bait density, reservoir density, fox

## Abstract

Fox rabies has been eliminated from vast areas of West and Central Europe, but cases still occur in the Balkans. Oral vaccination is an effective method for reducing the incidence of the disease in wildlife, but it requires monitoring if bait density is adequate for the density of the wildlife reservoir. We developed a methodology to evaluate the effectiveness of aerial vaccination campaigns conducted in Montenegro during autumn 2011 and spring 2012. The effectiveness of the vaccination campaign was assessed by (i) estimating the density of baits, (ii) estimating the distribution of the red fox, (iii) identifying critical areas of insufficient bait density by combining both variables. Although the two vaccination campaigns resulted in 45% and 47% of the country’s total area not reaching recommended density of 20 baits/km^2^, the consecutive delivery of both campaigns reduced these “gaps” to 6%. By combining bait and reservoir density data, we were able to show that bait density was lower than fox density in only 5% of Montenegro’s territory. The methodology described can be used for real-time evaluation of aerial vaccine delivery campaigns, to identify areas with insufficient bait densities.

## 1. Introduction and Aims of the Work

Rabies is one of the oldest diseases known in history [1]. The World Health Organization (WHO) estimates around 59,000 people die of rabies every year [2]. Most human cases are linked to infected dogs, that contribute up to 99% of all rabies virus transmissions to humans (https://www.who.int/news-room/fact-sheets/detail/rabies). Rabies cannot be treated; therefore, efforts must focus on prevention and control. Disease-prevention measures concentrate mainly on mammals that transmit the virus or on post exposure prophylaxis. Vaccination is the most effective tool for rabies prevention [1].

The incidence of rabies in humans has decreased dramatically in developed countries with the introduction of rabies vaccination of domestic animals, while canine rabies remains a problem in lesser developed countries [3]. In Europe the presence of wildlife reservoirs is a constant threat for potential reintroduction of rabies to susceptible dog populations. Oral rabies vaccination (ORV) is the most effective way to achieve a significant reduction in the number of cases in wildlife, [4]. With the application of ORV, red fox (*Vulpes vulpes*) rabies was eliminated from Western and Central Europe [5]. The first attempts of ORV were made in Switzerland during 1978 [6], followed by other European countries, e.g., Germany, France, and Belgium [7], reducing rabies cases in Europe from 21,000 during 1990, to 6000 during 2012. Anti-rabies vaccination became more effective once aerial distribution allowed a large scale and uniform supply of baits, with the first attempts at the end of the 1980s [8]. Over a period of three decades, vaccine baits have been distributed across nearly 1.9 million km^2^ in Europe, with nine previously endemic countries now fox rabies-free [9].

The efficacy of ORV, largely supported by EU financed projects, has resulted in the absence of rabies reports from the western Balkans countries since 2015, with the exception of a few cases in Serbia (source WAHIS website—www.oie.int/wahid and Rabies bulletin Europe—https://www.who-rabiesbulletin.org/).

During 2010, Montenegro started an EU-funded project: “Support for the control and eradication of rabies and classical swine fever in Montenegro”. The main purpose of the project was to financially and technically support local authorities in establishing the necessary capacity to adopt an aerial bait distribution approach in view of autonomously continue the vaccination programme until the ultimate elimination of rabies in wild carnivores. Hence, the breaking of the disease transmission chain from infected fox populations to domestic animals could be achieved as well as harmonization of the national legislative environment with the European requirements, to establish an effective control program. The above-mentioned programme is on-going.

In Montenegro, 86 rabies cases were recorded between 2010 and 2019, with the last cases reported during 2012 (as of 5 February 2020), 82.5% of which (71 of 86) reported in wildlife mainly in the red fox, and the remaining cases in domestic animals (source WAHIS website—www.oie.int/wahid and Rabies bulletin Europe—https://www.who-rabies-bulletin.org/).

A rabies elimination campaign should implement an evaluation of the success of the vaccination program. A retrospective analysis on the success of an ORV program highlighted that the main factors having a significant influence were (i) the proportion of a territory in each country ever affected by rabies and (ii) the area index value (being the area index an evaluation of the area covered with vaccination in different campaigns—varying from zero to one: campaigns that are entirely non-overlapping (index = 0) to those that entirely overlap and are equal in size to the maximum area ever vaccinated (index = 1)) [9]. This retrospective analysis was published during 2013 and took into consideration data from 24 European countries. Surprisingly, of all the previous evaluations of ORV in Europe only one descriptive study assessed territorial differences and factors contributing to ORV success, including bait density [10]. Specifically, the work of Mulatti et al. in 2011, implemented a geographical information system for the management of vaccine distribution, that proved to be useful, both for the planning and monitoring phases of the campaigns. Based on the guidelines of the European commission, vaccination campaigns need to guarantee a homogeneous distribution of baits between 20 and 30 baits/km^2^, depending on the red fox population density. Notwithstanding this guideline, there are usually no accurate data on reservoir density in the targeted areas and no evaluation on how the density and distribution of the reservoir could impact the ORV program.

Considering the above, one of the main points raised for a successful implementation of the project was the quantification of density and distribution of the main wildlife reservoir. Accurate information on red fox distribution in Montenegro was missing and was considered as a gap to be addressed for ensuring effectiveness of the vaccination campaign.

Nevertheless, the ORV program was successful with a consequent rapid decline in rabies cases in Montenegro. Henning et al. [11] pointed out some limitations suggesting that the efficacy of these vaccination campaigns was associated with landscape features.

In this paper we assess the efficacy of the vaccination campaigns during autumn 2011 and spring 2012 by (i) estimating the density of baits, (ii) estimating the distribution of the red fox, (iii) identifying critical areas of insufficient bait density by combining both variables.

## 2. Study Area Overview

The study area corresponds to the Republic of Montenegro. Montenegro is situated in the Western Balkans covering an area of almost 14,000 km^2^. The current human population is approximately 620,000 inhabitants. Montenegro is surrounded by Bosnia-Herzegovina, Serbia, Kosovo, Albania, and Croatia. The high diversity of geological features, landscape, climate, and location on the Balkan Peninsula, favor a high biological diversity. In fact, Montenegro is considered one of the most important hot spots of biodiversity in Europe. According to the Montenegrin Hunting Association and the National Park Administration, the fox population was estimated at approximately 10,000 animals in 2012. Domestic animals (mainly dogs and cats) are not systematically vaccinated against rabies. Stray dogs and cats cannot be adequately controlled, and rabies outbreaks continued to occur in urban areas before the implementation of the ORV campaigns. Until 2012, hunting restrictions were applied to areas where rabies occurred and in areas where the disease was registered within the past year. Here, the government organized a culling program for foxes (currently this measure is no longer applied).

## 3. Methods

### 3.1. Bait Delivery

Bait distribution was conducted using five airplanes, that operated for eight (autumn 2011) and six working days (spring 2012), respectively. The aerial distribution of baits was implemented in compliance with the criteria provided in the European Commission-Scientific Committee on Animal Health and Animal Welfare (EC-SCAHAW) reports [12]. The minimum targeted bait density was of 20 baits/km^2^, with urban areas and main water bodies exempted. The flight lines were designed to be at a maximum of 500 m (i.e., distance between two consecutive lines and from country national borders), whereas the distance between two consecutive baits was targeted to be 100 m. The automated bait vaccine distribution was connected to a geographical positioning system (GPS) device, and each bait was geo-referenced on the point of release and recorded and stored as a gpx file (GPS Exchange Format). Separate gpx files were produced for the 2011 and 2012 campaigns, by each airplane and for each flight.

### 3.2. Fox Monitoring and Transect Fox Density

Fox density was estimated during two census campaigns, carried out nation-wide during summer 2011 (from July to October), and spring 2012 (from March to April). For each period a different monitoring method and a different network of transects were used, according to the distinct environmental conditions. Specifically, considering that the vegetation cover in summer does not allow the application of count methods based on direct observation of animals, an indirect method was used in summer and direct methods in spring. Monitoring was performed by local hunters who received specific training.

Indirect monitoring: considering the dense vegetation cover, in summer we applied an indirect census method using the fecal pellet count (FPC) along transects (i.e., each transect with a minimum length of 10 km) [13]. Transect locations were stratified based on land use cover to homogenously sample all habitat types. Specific transect locations (i.e., the starting point of the transect) were randomly placed inside each land cover category. The final monitoring effort was of one transect every 15,000 ha of territory. The transects were geo-referenced with GPS instruments and the operators (i.e., local hunters) monitored the same transect twice. Firstly, all the feces were counted and removed (i.e., transect cleaning) and, after a month, the observers took note of the new feces detected along the transect. Geographical coordinates of the feces location were recorded with a GPS device. A density value (i.e., number of animal/km^2^) was derived with data from the second examination, using the formula provided below [14].
*Transect fox density = (number of feces found/km^2^)/(defecation rate × days between repetitions)*(1)

Direct monitoring: during spring monitoring, we applied a direct census technique, using a spotlight night count along transects randomly placed inside each land cover category (i.e., each transect with a minimum length of 10 km). This technique is one of the most effective methods in wildlife monitoring [15]. The basic assumption of the method is that animals are more detectable at night as they are more active. The combination of the classical method with the “Distance” protocol developed by Buckland et al. [16], and applied in the current study, allowed to increase its precision and accuracy. The Distance method corrects estimations based on detection probability of the animals as a function of distance from the observers. In particular, detection probability is considered equal to 1 on the transect line and decreases with increasing distance from it. The distribution of the observed animals and their distance from the transect are used to build a function that describes the probability of detecting an object at a given distance. Modelling the reduction of detectability with distance from the transect allows estimating the total number of animals in the area investigated, based on the number of observations. Spotlight counts were conducted by a team of two observers (i.e., one driving the vehicle and the other one counting the animals) in a vehicle driven at a speed of 5–10 km/h. Both transects and foxes observed were geo-referenced with a GPS. Perpendicular distance of each fox from the road was measured with the use of a laser telemeter (Leica—Rangemaster 900).

Data collected were introduced in Distance 6.0^®^ [17]. The Distance software allows estimation of the population density, based on the observation made on each transect, applying the protocol described above. To obtain the density value (foxes/km^2^), a half-normal function with cosine expansion was implemented [16]. This function is considered to most likely represent field data that were collected under well-controlled conditions and was the one that best fitted the observed data.

### 3.3. Identification of Critical Areas of Insufficient Bait Density

Data collected in the field were used to generate spatial layers using Quantum GIS 1.7.4 [18]. A common 1 km^2^ vector grid for bait and fox density, covering the national territory, was created using a QGIS plug-in (“create vector grid”).

(a) Bait density analysis

Bait density analysis was conducted using background information to identify the sample areas. Information from the Corine Land Cover Project was used to obtain data on land use and to identify exempt areas (i.e., urban areas and main water bodies) (http://www.eea.europa.eu/publications/COR0-landcover). Corine Land Cover data were used as vector shapefile at a scale of 1:100,000. Moreover, the Open street map project data were used to determine national boundaries (http://www.openstreetmap.org/#map=5/51.500/-0.100).

Bait density was calculated by overlaying the 1 km^2^ vector grid and the point layers of the georeferenced baits. The QGIS plug-in “count points in polygon” allowed to automatically count the number of baits in each grid square.

(b) Fox density analysis

Fox density was calculated by overlaying the 1 km^2^ vector grid with the density estimated for each transect. In unsampled cells, spatial interpolation (i.e., using the Ordinary Kriging analysis) was implemented. Kriging is a method that uses a limited set of sampled data points to estimate the value of a variable over a continuous spatial field. The Ordinary Kriging approach assumes the stationarity of data (mean and variance of the values is constant across the spatial field).

Normality of the distribution of the density estimates was evaluated with the Shapiro–Wilk test. The non-parametric Wilcoxon test was used to compare the density estimates between both years. A *p*-value of <0.05 was considered statistically significant.

(c) Critical areas of insufficient bait density

A risk map was built to quantitatively and visually evaluate the relationship between bait density and host density, to identify the presence of “critical areas”. Critical areas were defined in our study as zones with the densities of distributed baits lower than the estimated density of the reservoir. In other words, we focused our attention on the squares where the number of baits was below a ratio of 1:1 with the estimated number of targeted animals. In dogs, if the safe threshold to guarantee a proper vaccination coverage is considered 70% of the susceptible population, in our study we used a higher threshold, considering the possibility that some baits were lost during the distribution process.

The software R was used for all statistical analyses [19].

## 4. Results

(a) Bait density analysis

The total area covered during the aerial distribution, excluding water and urban areas, was 12,842 km^2^ (calculated on a 1 km^2^ grid). During the two vaccination campaigns, 274,219 baits were distributed during Autumn 2011 and 273,888 baits during Spring 2012, with a mean bait density of 19.94 baits/km^2^ during 2011 and 20.04 baits/km^2^ during 2012. Bait density differed significantly between the two campaigns. Comparing bait frequency density for each square of the grid, each sampling unit received a highly different number of baits during the first and second distribution campaign (Wilcoxon signed rank test with continuity correction; V = 32,963,824, *p*-value = 1.532 × 10^5^). This difference was mainly due to the attempt to adjust the distribution of baits during the second distribution session, taking into consideration the results of the first one.

Descriptive statistics of bait density at national level is reported in Table 1.

(b) Fox density analysis

Foxes were monitored, respectively, along 67 (indirect census) and 44 (direct census) transects. The overall length of the 111 transects was 1337 km, covering the whole national territory and all types of land use cover (Figure 1).

Fox density values estimated with census activities are reported in Table 2.

There are no statistically significant differences between the results of the direct and indirect censuses (Wilcoxon rank sum test with continuity correction; W = 1620.5, *p*-value = 0.3786).

Data from the two census campaigns were considered comparable and congruent, and it was possible to interpolate them together through the Ordinary Kriging analysis.

Kriging results highlighted the presence of areas with very different reservoir density (Table 3). The majority of Montenegro (58%) had an estimated fox density between two and four individuals per km^2^.

Estimated fox density and monitoring network (transects) are reported in Figure 2.

(c) Critical areas of insufficient bait density

During 2011, 1919 km^2^ (5.4% of the national territory) were classified as “at risk” with bait density equal or lower than host density. During 2012, it lowered to 1836 km^2^ (4.5% of the national territory). In both campaigns these areas were in the eastern and western parts of the country (Figure 3).

## 5. Discussion

The current study provides an evaluation of the efficacy of aerial bait distribution for the prevention and control of wildlife rabies, describing an analytical approach that can be used to evaluate in real time the efficacy and gaps of a vaccination campaign. The proposed analysis allowed not only to describe the gap in the vaccination campaigns but also to adapt the vaccination strategy, identifying the critical areas with potential vaccination coverage problems and consequently achieving successful elimination of the disease (no more rabies cases have been registered after 2012).

Bait and reservoir density are just two factors that influence the success of a rabies control plan but while there is a substantial agreement on the minimum bait density value that influences vaccination success, data about reservoir density are lacking. Bait density monitoring is very important to evaluate both the minimum density value of baits/km^2^ and the presence of critical areas with low vaccination coverage or an imbalance between bait and reservoir species density.

Alternatively, wildlife monitoring is of pivotal importance to assess and localize the presence of areas with a high density of the main reservoir (i.e., fox) and other possible competitors for bait intake (i.e., other wild carnivores, omnivores, and dogs) and for quantifying interactions between different reservoirs.

Even if a clear relationship between disease prevalence and host population density currently does not exist for rabies [20], it has been clearly demonstrated that the performance of vaccination programs is influenced among other factors by the reservoir density [21]. The approach used in EU projects, with an identified target bait density, works in most situations but can demonstrate limitations in areas with high fox densities and poor vaccination coverage [22]. For this reason, the approach proposed in our study can be particularly useful to identify and address gaps (areas with persistent rabies cases) during the course of vaccination campaigns.

Few ORV programmes take into account both bait and reservoir density [10,23,24], and none has tried to merge baits and reservoir information to identify post vaccination critical areas. The main constraints are related to the lack of georeferenced data and background information. This type of analysis, complex from an analytical point of view, is made possible by the integrated use of GPS data, GIS software, and geodatabase that provides environmental information [25].

GIS has proved to be an efficient tool in epidemiological investigations and wildlife disease management. Spatial visualization allows to rapidly interpret information, with consequent optimization of control programmes [26]. The first documented use of GPS/GIS analysis in rabies control dates to 1991 [25] during a Wyoming skunk rabies epizootic. Before this date, GIS tools were used during vaccination campaigns to map and select vaccination sites [24], to improve surveillance data [23], to map the spatial distribution of rabies [27], or to develop a GIS based real time internet mapping tool for rabies surveillance [28]. In our study, the implementation of GIS for the management of vaccine distribution proved to be useful to monitor the bait distribution quality in nearly real time.

Previous studies demonstrated the usefulness of a GIS approach to identify gaps [10]. Starting from this basis we evaluated in real time the coverage of a vaccination campaign to apply corrective measures. The analysis demonstrated some problems in bait distribution, underlying the presence of critical areas where the minimum density required by European standards was not reached. Comparing the extent of areas without sufficient bait density in our study, to one [10] in Italy (5.9% to 6.1%), the vaccination campaign in Montenegro was characterized with a higher percentage of areas covered unsatisfactorily (45.0–47.3%). Moreover, in Montenegro an additional problem was represented by areas with a total number of baits much higher than required (i.e., up to 106 baits/km^2^), with animals potentially receiving vaccination doses higher than needed.

However, it is difficult to compare this study with the one performed by Mulatti et al. [10] due to the different mean of distribution. In Montenegro, baits were distributed by airplane whereas in Italy by helicopter. Indeed, airplane distribution is less precise but less expensive and more appropriate for country-wide coverage.

The fox monitoring provided valuable results. The monitoring was homogeneous overall and transects were randomly distributed in the different land cover categories, thus allowing information on fox density in different geographic and environmental conditions. The use of two census techniques (i.e., based on different theoretical assumptions) allowed to cross validate the census information and to work with precise and accurate data. Further, the Kriging interpolation used to evaluate density in unsampled areas was very useful for modelling fox distribution throughout the country, allowing a well-balanced cost/benefit approach. Further improvements on the interpolation method should consider the evaluation of the uncertainty of the estimation in each area and the use of appropriate covariates (e.g., environmental characteristics) that could influence density estimation locally. Finally, the training provided improved the detection capabilities of all hunters involved in the monitoring activities and ensured promising results for the long-term sustainability of the monitoring operations.

Mean fox density in Montenegro (3.65 fox/km^2^) is similar with the values found in other countries. In particular, fox density (i.e., fox/km^2^) ranging between 0.21 and 2.23 has been described in Britain [29,30], between 1.02 and 2.1 in Poland [31,32], between 1.2 and 3.0 in Central Victoria [33], and up to 4.3 in New South Wales. Unfortunately, for most of the Balkan countries it is difficult to find published data on fox density. Fox density estimation is a suggested key component in planning an ORV campaign.

In this study we provided a novel approach to assess ORV campaign. A complex approach is implemented by merging results of bait and fox density analysis. Even if the application of spatial analysis techniques requires specific expertise, once the system to perform this kind of analysis is put in place, and know how is correctly transferred, then it can be run on a routine basis. This allowed to draw clearly delimited risk zones, and to identify critical areas with bait density equal or lower than host density. Critical areas were mainly aggregated in two clusters in the eastern and western parts of the country, characterized by very high reservoir density. This is probably due to the presence of particularly favorable environmental conditions (i.e., forested areas in the eastern part of the country and mixed farming areas in the western part).

A fox sampling scheme to assess the ORV campaigns effectiveness was completed for the first year of monitoring with satisfactory results. Based on the results of the analysis, 80% of samples were positive for vaccine uptake and 46% had evidence of rabies virus neutralizing antibodies (VNA) [34]. In the other countries covered by the IPA Western Balkans Project, the bait uptake ranged between 28% and 70% and the VNA levels between 7% and 33% [34]. These results highlight the effectiveness of real time monitoring of vaccination coverage. These data are not easy to obtain due to several challenges such as (i) difficulties of obtaining permission in allowing fox hunting for rabies surveillance in protected areas and hunting outside of the stated hunting season; (ii) unexpected environmental constrains (specifically during summer 2012, there were large scale forest fires and high temperatures); (iii) difficulties in relation to the disposal of animal by-products (baits).

Our analysis highlighted some critical points to be considered in planning a vaccination campaign. The ORV programme should be conducted considering the reservoir density and the relationship between bait and reservoir density, that lead to potential gaps in vaccination coverage. These indicators can be used to correct the minimum bait density proposed by European guidelines, to address specific problems in areas where rabies persists and cases continue to occur.

This study provides a relevant strategy in the evaluation of fox rabies vaccination campaigns. The methodology presented has significant benefits in terms of improving the planning and implementation of an ORV campaign. Bait density is one of the primary factors affecting the cost and effectiveness of ORV campaigns. Distribution of higher densities of baits in areas where reservoir densities are known to be high may be warranted to ensure sufficient baits are available. At the same time an overdistribution of baits in low density areas should be avoided.

The approach described in this manuscript is useful for planning and iterating vaccination campaigns and could complement other measures of campaign efficacy, such as monitoring postvaccination coverage using vaccine markers, as well as surveillance of rabies prevalence in the reservoir species.

Additional studies, in line with the one carried out by Henning et al. [11] to quantify bait uptake and vaccination success (i.e., evaluation of seroprevalence and localization of vaccinated foxes) are recommended, especially in the critical areas. These studies may yield valuable insights into long-term effects on reservoir immunity. Ideally, in addition to the analysis presented here, information on estimated vaccination coverage achieved by area would help to validate the predictions of population density as a basis for guiding bait distribution.

As of 2020, rabies is not eliminated from the Western Balkan region yet. One EU funded project is expected to be launched during the first quarter of 2020 in Albania. The project is based on the outputs from the meeting of the Standing group of Experts on Rabies in Europe (GF-TAD) held during February 2019 where the goal to accelerate the final elimination of rabies has been pointed out, in line with the EU objective (by 2020) [35]. Therefore, it was decided that the vaccinations against rabies in Albania will be continued for an additional three aerial bait vaccination campaigns (2020–2021). Further, an additional EU funded regional multi-country project is close to its inception involving Albania, Bosnia and Herzegovina, North Macedonia, Montenegro, Serbia, and Kosovo. The project addresses the control of transboundary diseases, including the final elimination of rabies in the region within the next few years.

Although the primary data presented in this manuscript were collected 8 years ago, the objective of this communication was to develop and describe a methodology that can be used for real-time evaluation of aerial vaccine delivery campaigns to identify areas with insufficient bait densities. Thus, the approach described here is highly applicable for the evaluation of any future vaccination campaigns, to improve the success of the future ORV in the region.

## Figures and Tables

**Figure 1 tropicalmed-05-00032-f001:**
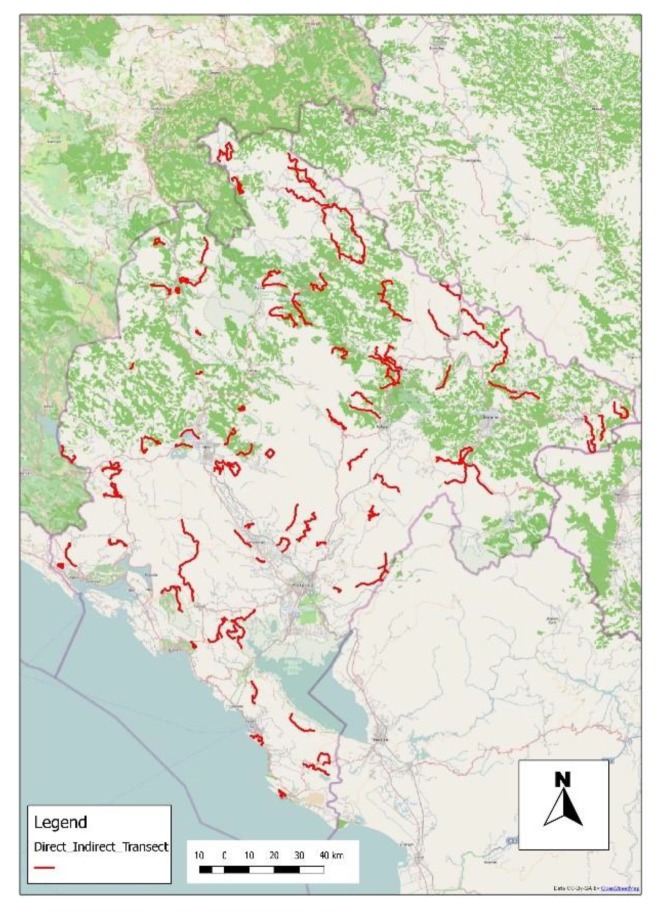
Montenegro wildlife monitoring net during 2011 and 2012. The monitoring transects are represented with a red line.

**Figure 2 tropicalmed-05-00032-f002:**
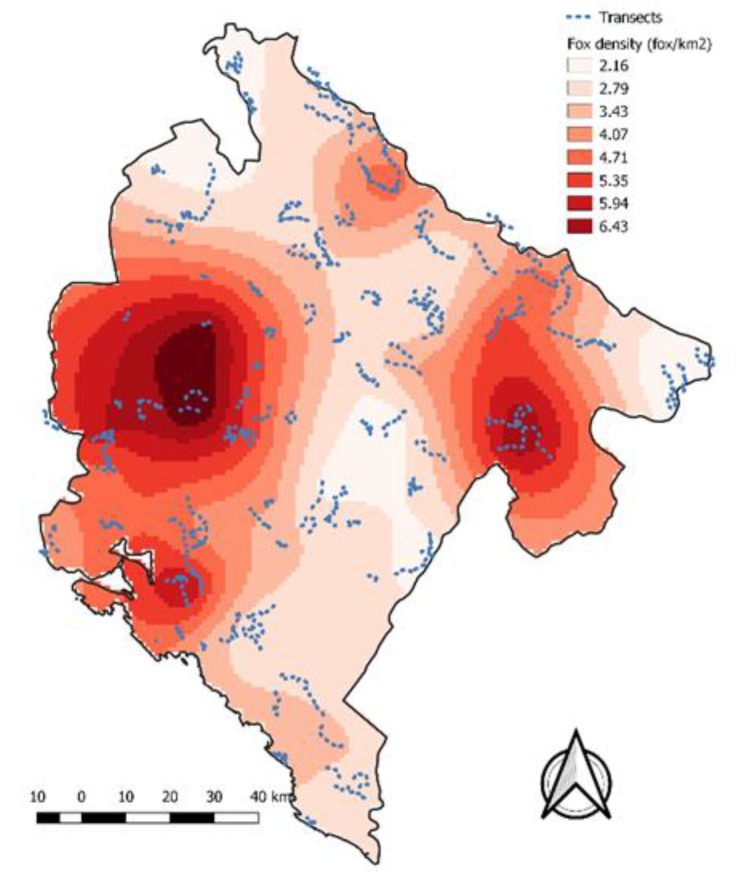
Fox density calculation through Ordinary Kriging interpolation and monitoring network.

**Figure 3 tropicalmed-05-00032-f003:**
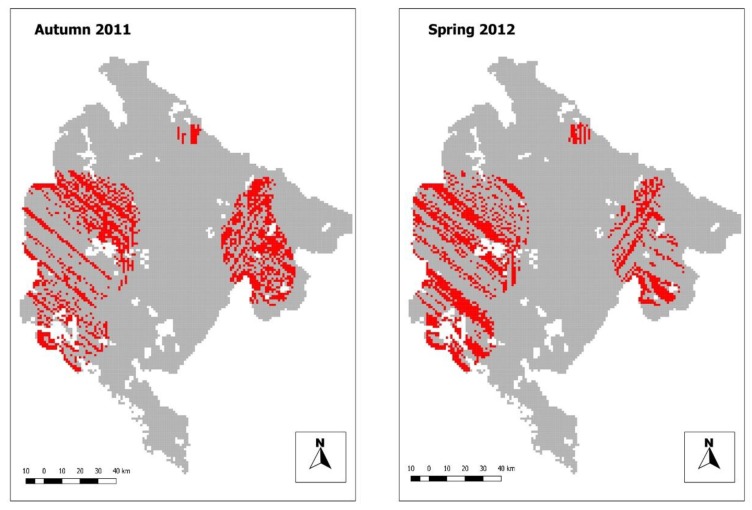
Critical areas: autumn 2011 campaign (on the left), spring 2012 campaign (on the right).

**Table 1 tropicalmed-05-00032-t001:** Descriptive statistics for estimated rabies bait density (number of baits/km^2^) in Montenegro. The table report the mean bait density with 95% confidence interval, and the percentage of area covered, considering different bait density classes.

Bait Density	Autumn 2011	Spring 2012	Combined 2011 and 2012
Mean (IC 95%)	19.94 (19.78–20.10)	20.04 (19.89–20.18)	39.98 (39.74–40.21)
Coverage (lower than 20 baits/km^2^)	47.3%	45.02%	6.03%
Coverage (lower than 15 baits/km^2^)	20.3%	17.2%	4.8%
Coverage (lower than 10 baits/km^2^)	10.4%	7.3%	3.9%
Coverage (lower than 5 baits/km^2^)	5.9%	4.8%	3.3%

Only 3% of the national territory did not receive any bait in either of the two campaigns.

**Table 2 tropicalmed-05-00032-t002:** Fox density (number of animals/km^2^) estimated with direct and indirect monitoring during 2011 and 2012.

Density	Indirect Census	Direct Census	Overall from Both Methods
**Minimum**	0	0	0
**Maximum**	14.30	11.20	14.30
**Mean**	3.57	3.77	3.65
**Median**	2.30	3.25	3.0
**SD**	2.87	2.58	2.75

**Table 3 tropicalmed-05-00032-t003:** Classification of Montenegro in areas with different fox densities. Each area is indicated with its extension in km^2^ and its percentage in relation to the entire Montenegro area.

Fox Density	km^2^	Percentage
0 to 2	725	5.6
2 to 4	7502	58.4
4 to 6	3976	31.0
More than 6	639	5.0%

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
