# Peer review of "Wildlife and Bait Density Monitoring to Describe the Effectiveness of a Rabies Vaccination Program in Foxes"

_tropicalmed, 2020, doi:10.3390/tropicalmed5010032_

Round 1
Reviewer 1 Report
Paolo and co-authors used GIS based techniques to predict areas where oral rabies vaccine (ORV) distribution was less than the estimated population of the target (foxes). I have no major concerns with this manuscript. English language editing is needed throughout.
Minor Comments:
The introduction seems overly broad and could be shortened.
Methods
l. 128-147: the two different methods are described as SA for summer(?) and SP for spring (also referred to as winter in l. 134?) but later in the manuscript direct and indirect census are used. To avoid confusion I suggest eliminating the SA and SP and use direct and indirect census.
l. 174-178: I would suggest integrating these bullet points into the text.
l. 197: Justification of use of a 1:1 ratio is required. For dogs it is accepted that 70% heard immunity is sufficient to disrupt transmission of canine variant. I don't know if similar calculations have been done for red foxes. I know that red foxes are more sensitive to rabies virus infection than dogs so it could be argued that 1:1 ratio is needed.
Results
Table 2 and 3 formatting should be similar (why is table 2 so much larger than table 3?)
l. 249-252: I think it is important to emphasize that even though rabies cases in foxes decreased, cases continued to be detected (l. 57-58). The co-authors argument would be strengthened if those cases could be traced to areas of insufficient bait density. If cases can be traced to a region, a different color dots could be overlaid on fig. 4 to show the overlap of cases and insufficient bait density.
Discussion
Again emphasizing that this additional analysis is needed to eliminate rabies in the target population is important. Most readers will assume that traditional evaluation of baits/km2 is sufficient since it has been successfully used in other countries to eliminate rabies. The co-authors need to advocate more forcefully for the benefits of their analysis.
l. 316: what is the comparable number for Montenegro?
Table 4: It's not clear what the range represents (average plus/minus standard deviation?).
l. 326-331: Again emphasize that minimum bait distribution was met but because of uneven distribution of baits and target population rabies elimination was not achieved. This supports the argument that minimum bait distribution guidelines should be reevaluated.
l. 332-337: Do the co-authors propose supplemental ORV distribution in areas of insufficient bait density? A follow up study would be very interesting if it could be shown that rabies was eliminated based on the co-author’s analysis.
Reviewer 2 Report
This study developed a new real-time evaluation method to evaluate the effectiveness of oral rabies vaccination programs in Montenegro in the fall of 2011 and the spring of 2012. There are three items to assess the effectiveness of vaccination: (1) estimating the density of baits distributed; (2) estimating the distribution of the Red fox; (3) identifying critical areas of insufficient bait density by combining both variables.
Some questions:
Line 72: At the end of this line, there is only one “i” but no “ii” … and others. Line 136: what is the abbreviation of SA? Line 137: FCP is the correct abbreviation of faecal pellet count? Line 147: what is the abbreviation of SP?Author Response
Please see the attachment

Reviewer 3 Report
The study describes evaluation of an oral vaccination programme using comparison between bait coverage and estimates of fox population distribution. This seems like a logical and rational approach to national campaign monitoring in near-real time, which would be of relevance to ongoing ORV areal distribution efforts. I have some general and specific comments.
General comments
The introduction combines comments on the impact and control of canine transmitted rabies with that of sylvatic rabies, which creates confusion about the relevance of mass vaccination of the canine and wildlife population respectively. The issue of rabies and approach to its control is totally different in regions affected by canine versus sylvatic rabies and so a clearer distinction should be made when introducing the situation to the reader (please see specific comments below).
An immediate question raised when reading the manuscript is the long period of time since the campaigns in the study, which were 8 years ago, and how these are relevant to the situation today. I think it would be important to give context as to how these relate to progression of control efforts in the region since then and where these results would be of benefit if applied. Have these campaigns continued in Montenegro in the same format since the study period? Where else are vaccination campaigns currently using the same approach to areal bait distribution and could benefit from these methods? Are there areas of Europe which appear to be failing to effectively control the disease with existing EU recommended bait densities and could benefit from this more refined approach? Is it possible to give any indication of the degree of wastage due to over vaccination of regions with low reservoir densities when the blanket EU recommended density is applied?
Ideally in addition to the analysis presented here, information on estimated vaccination coverage achieved by area would help to validate the predictions of population density as a basis for guiding bait distribution. Although this approach may be useful in planning and iterating campaign structure, more emphasis should be made in the discussion about the importance of other measures of campaign efficacy, particularly monitoring post-vaccination coverage using vaccine markers, as well as good surveillance systems to monitor rabies incidence in the reservoir species.
The wording and structure of the text needs attention in a number of places (please see specific comments).
Specific comments:
Line 32 - 34 – The current study relates to control of rabies in the red fox population, however context is not given to the number of human deaths referenced here, virtually all of which are a consequence of canine transmitted rabies. As it currently reads it is confusing to understand the significance and context of sylvatic versus canine transmitted rabies.
Line 33 - Reference [2] is outdated from 1993, there are more current studies on human rabies disease burden, e.g. Hampson et al 2015.
Line 39 - 40 – Reference [3] is specific to the USA, however the text suggests that the incidence of human rabies has decreased dramatically globally, for which evidence is not given.
Line 39 – 46 – The distinction between sylvatic rabies and the role played by dogs is not clearly stated here. It would be useful to communicate the distinction between settings where roaming dog populations are the primary reservoir and therefore mass dog vaccination is performed to achieve elimination as opposed to areas where a wildlife species is the primary reservoir and therefore are targeted for vaccination, but domestic dogs are still vaccinated to reduce risk to people. At the moment it is confusing as to the role played by dogs and for what reason they are vaccinated in the context of sylvatic rabies.
Line 40 - 41 – Again the context here seems confusing, emphasising the importance of vaccinating dogs in regions, where perhaps there is not a free-roaming dog population and other species are the primary reservoir.
Line 43 – 46 – sentence does not read well, suggest re-writing.
Line 54 – 56 – Please add a reference to support this statement.
Line 57 – 60 – Sentence does not read well, please reword.
Line 50 – 51 – What year was aerial distribution introduced?
Line 54 – 65 – It is quite difficult to follow both the geographic and chronological progression of activities in Europe, with the time periods jumping around a lot. For example: Line 55 “…Balkans countries since 2015…”, Line 59 “Montenegro… between 2010 and 2019”, Line 61 “In 2010 in Montenegro”, Line 63 “completed in April 2013”, Line 68 “in autumn 2011 and spring 2012”. Suggest re-writing with a chronological flow to help the reader follow.
Line 66 – 67 – repetition of Line 50 – 51.
Line 68 – 69 – This reference to EU funding would be better combined with previous reference to EU funding (Line 61). Here it is also confusing as to who funded as opposed to implemented the campaigns. The sentence states that the first two campaigns were funded by the EU, but then subsequently implemented by the Montenegrin government. Did the EU also implement the first two campaigns or just fund them? If so, who implemented those campaigns? Did the Montenegrin government both fund and implement the campaigns thereafter?
Figure 1 – It would be useful to include the geographic extent of the regional bait distribution campaigns in the surrounding countries at the time of study, which would give context as to how the efforts described in Montenegro fitted into the wider effort. At present it is not clear if Montenegro was a border country in the EU-funded campaigns or surrounded by countries also conducting similar control efforts. As it stands the figure communicates limited additional information about the study other than the location of Montenegro, which probably does not warrant a figure by itself.
Line 129 – 130 – Were exactly the same transects used by both the direct and indirect survey methods? I.e. were surveys repeated along the same transects using the two methods or different transects were used for each method? This is not currently clear.
Line 136 and Line 147 – It’s not immediately clear what the SA and SP abbreviations in the subtitles refer to. They are referenced in Line 130, but then not used again in the manuscript. This seems to create unnecessary extra effort for the reader to work out what the subtitles refer to. Please change the subtitles to be more explicit.
Line 138 – “Transects locations…” suggest change to “Transect locations…”
Line 139 – “…stratified based on land use cover, in order to homogenously sample all different type of habitats…” suggest rewording to “…stratified based on land use cover to homogenously sample all habitat types…”
Line 138 – 140 – How were specific transect locations (e.g. starting points) selected? Was this somehow randomised?
Line 158 – 160 – As for the summer counts, how were the transect starting points determined? Were these randomly selected? Please include the method of transect location selection in the methods.
Line 184 – Was any reference data from unsampled regions used as the basis for extrapolation of population density estimates? Or does Kringing interpolation purely infer differences by region based on the distribution of the survey data? As mentioned in the discussion, population density is likely to be highly correlated by land type and so estimating population densities by land type and then using this as the basis for extrapolation would seem to be the most reliable way of estimating population distribution.
Line 195 – 198 – Sentence does not read well, please re-write to clarify.
Line 204 – “Bait density different significantly…” change to “Bait density differed significantly…”
Line 207 – remove “.” before parentheses.
Line 210 – 211 – does not make sense as a stand-alone paragraph – does this refer to the information in the previous paragraph? If so, combine and re-word sentences to clarify.
Line 212 – 214 – These headline results would work better as the second sentence of the results, after the total area covered. The figures about bait distribution density presented here should also be combined with the other statements about bait density (Line 205 – 207) as opposed to being presented separately later which disrupts the flow and clarity.
Table 1 – Descriptive stats of the two campaigns could be presented more concisely without the need for separate rows for min, max etc. Perhaps the table could summarise the mean bait density for each campaign with 95% confidence intervals, as well as the data presented in the text of Lines 217 – 220.
Line 220 – 223 – This information is important, but could be re-worded for clarity. The use of parentheses here makes the sentence disjointed. Suggest rewording.
Figure 3 – This figure would be more appropriate in Supporting Materials as opposed to in the main manuscript as there is redundant and duplicated information. A single final figure overlaying the transect GPS points with the final population density model would be more concise and impactful (i.e. suggest combining the maps presented in Step 2 and Step 3 into a single plot).
Line 249 – 251 – It seems that the critical area results are first given as percentages for each campaign and then later repeated in a separate sentence for km2. This would be more concise if combined, perhaps giving the percentage in parentheses.
Line 263 – “an unbalanced ration baits / reservoir.” Does not read well. Perhaps “… an imbalance between bait and reservoir species density.”
Line 266 – “…(others wild carnivores…” change to “…(other wild carnivores…”
Line 268 – “Even if currently a clear relationship…” rewording needed. “Even if a clear relationship between disease … … does not currently exist for rabies…”
Line 270 – Please give references for where the performance of vaccination campaigns is influenced by reservoir species density.
Line 271 – “…works in most of the situation but…” reword, perhaps “…works in most situations…”
Line 272 – reword “works in most of the situation but can demonstrate limits in case of areas with high fox densities and reduce vaccination distribution”
Line 277 – should be “constraints”
Line 284 – should be “tools”
Line 285 – should be “campaigns”
Line 288 – “…usefulness to use a…” needs rewording
Line 289 – “Our analysis started from this basis to evaluate in real time the coverage of vaccination campaign in order to apply corrective measures.” Does not read well
Line 296 – 297 – needs to be integrated with a relevant paragraph rather than being a single sentence in a paragraph of its own.
Line 302 – 303 – as for above, this sentence should be integrated with the paragraph on use of GIS (Lines 281 -287)
Line 304 – 305 – Again please include here how transects were selected to ensure random distribution if used. It is still possible for sampling to be homogenous, and yet biased if transect location selection was not randomised.
Table 4 – Inclusion of such specific data in tabular form in the discussion does not seem relevant to the current study or ongoing bait distribution campaigns. It could be included in the text, although comment should be made about how this applies to areas of ongoing bait distribution in Europe.
319 – 320 – The method is described as “complex” and so I would be concerned for how this method be easily replicated in a cost-effective way to bait distribution programmes elsewhere?
Line 322 – 323 – It would be relevant to comment on whether ongoing rabies cases occurred in these areas and also whether vaccination coverage was evaluated. It may be that surveillance data or vaccination coverage information is incomplete, however it would be important to comment on whether this data is available for the study period or not and perhaps the challenges in gathering this most critical information to campaign monitoring.
Round 2
Reviewer 3 Report
Thank you to the authors for their revisions based on the review, following which the manuscript has been improved considerably. I now believe that the manuscript is suitable for publication. Congratulations to the authors on this relevant and useful study.
Attention to sentence wording and readability is still needed in some places. For example:
Line 40 - 42 - revisit wording
Line 311 - "...is still difficult to found published data..." should be "find".
Line 315 - 316 "...spatial analysis requires a quite specific knowledge..."
Line 355 - 357 wording
Line 363 - 365 wording
Is there a reference to a report or Institution for the percentage results reported on vaccine uptake (line 322 - 324)? The same for Line 326 - 328.
